# Inactivation of Ebola Virus and SARS-CoV-2 in Cell Culture Supernatants and Cell Pellets by Gamma Irradiation

**DOI:** 10.3390/v15010043

**Published:** 2022-12-23

**Authors:** RuthMabel Boytz, Scott Seitz, Emily Gaudiano, J. J. Patten, Patrick T. Keiser, John H. Connor, Arlene H. Sharpe, Robert A. Davey

**Affiliations:** 1Department of Microbiology, National Emerging Infectious Diseases Laboratories, Boston University, Boston, MA 02115, USA; 2Department of Immunology, Harvard Medical School, Boston, MA 02115, USA

**Keywords:** Ebola virus, SARS-CoV-2, inactivation, sterilization, filovirus, coronavirus, radiation treatment

## Abstract

Viral pathogens with the potential to cause widespread disruption to human health and society continue to emerge or re-emerge around the world. Research on such viruses often involves high biocontainment laboratories (BSL3 or BSL4), but the development of diagnostics, vaccines and therapeutics often uses assays that are best performed at lower biocontainment. Reliable inactivation is necessary to allow removal of materials to these spaces and to ensure personnel safety. Here, we validate the use of gamma irradiation to inactivate culture supernatants and pellets of cells infected with a representative member of the Filovirus and Coronavirus families. We show that supernatants and cell pellets containing SARS-CoV-2 are readily inactivated with 1.9 MRad, while Ebola virus requires higher doses of 2.6 MRad for supernatants and 3.8 MRad for pellets. While these doses of radiation inactivate viruses, proinflammatory cytokines that are common markers of virus infection are still detected with low losses. The doses required for virus inactivation of supernatants are in line with previously reported values, but the inactivation of cell pellets has not been previously reported and enables new approaches for analysis of protein-based host responses to infection.

## 1. Introduction

Since the 1940s, the rate of emergence of novel infectious diseases has steadily increased. Of the newly emerging diseases, a majority are zoonotic in origin and a significant portion are viruses [1,2]. Notable examples include Machupo virus in 1959, Ebola virus in 1976, H5N1 bird flu in 1996 [3], SARS-CoV in 2002 [2,4] and SARS-CoV-2 at the end of 2019 [5]. A variety of factors including host and pathogen characteristics, as well as environmental and societal changes, have been linked to the emergence and spread of zoonotic pathogens [6,7]. SARS-CoV-2 is but the latest emergent viral pathogen to threaten human health and cause widespread disruption of national and global economies. The ongoing climate crisis and associated impacts on population distribution and migration are projected to increase the number of people threatened by newly emerging and re-emerging infectious diseases, and to continue to foster conditions that are conducive to zoonotic pathogen emergence [8].

There is a clear need for research on all aspects of emerging zoonotic viruses, from studies on the ecological bases of emergence to immunological investigations and basic research on the viruses themselves. Many emerging viruses can only be studied at high biocontainment, with the most pathogenic requiring biosafety level-4 (BSL4). Most high containment labs are equipped with only the most necessary equipment required for working safely with these pathogens. Performing work with live viruses also poses hazards to personnel as it comes with risk of exposure. It is therefore important, where possible, to inactivate virus-containing material to eliminate the biological hazard, allowing analytical work to be performed outside of high biocontainment. The ability to inactivate samples and remove them from containment enhances the analytical power that can be brought to bear on emerging disease questions.

Common methods of inactivation include aldehyde-based fixation, a combination of heat and denaturation, UV light, chemical disruption and gamma radiation [9,10,11,12,13]. Gamma radiation penetrates deeply into all materials resulting in radiation-based ionization that permanently breaks chemical bonds in biological materials. Additionally, ionization of water produces free radicals that can in turn modify chemical groups found in RNA, DNA, proteins, and lipids. Overall, this action denatures and cleaves proteins, lipids, DNA and RNA rendering them biologically inactive [14]. Since virus infection requires activity of each of these macromolecules (structural proteins and lipids) and contiguous stretches of DNA or RNA in the virus genome, the virus is rendered non-infectious [15].

Here, we build on previous reports that have shown gamma radiation as an effective inactivation treatment for coronaviruses, filoviruses, arenaviruses [11] and other virus types [13,15]. Our study tested gamma irradiation as an inactivation method for culture medium containing SARS-CoV-2 and Ebola virus (EBOV) and cell pellets infected with these viruses. Other coronaviruses, such as SARS-CoV and MERS-CoV have been reported to be relatively easy to inactivate [16]. In contrast, filoviruses, which include EBOV, require relatively higher doses of radiation for inactivation [11,12]. Aside from significant morphological differences, each virus type has a similar composition, being composed of an RNA genome bound by proteins that stabilize the RNA and form a nucleocapsid, which is covered by a lipid bilayer in which the glycoprotein spikes are embedded. We used recombinant forms of each virus type that have an extra gene encoding green fluorescent protein (GFP) to serve as a sensitive marker of infection to compare how each is inactivated by gamma radiation. We extend previous studies by evaluating the inactivation of virus present in infected cell pellets and also measure the impact of treatment on the detection of cytokines produced by infected cells which are often used as markers of virus infection [17]. Cytokine activity is also a useful readout of macromolecular viability for downstream applications such as measuring immune responses or proteomic work.

## 2. Materials and Methods

### 2.1. Cells and Viruses

Cells used for this study were African green monkey kidney cells Vero cells clone E6 (ATCC, Manassas, VA, USA). These cells have been shown in multiple studies to be permissive to both Ebola virus (EBOV) and SARS-CoV-2 [18,19]. Specific virus strains used were icSARS-CoV-2, derived from SARS-CoV-2/human/USA/WA-CDC-WA1/2020 (BEI Resources; GenBank MT020880), a recombinant virus that encodes mNeonGreen and produced by Dr. Shi, UTMB, Galveston, TX, USA [20], and Zaire Ebolavirus strain Mayinga (GFP) subtype Zaire, a recombinant virus clone that encodes enhanced green fluorescent protein (GFP) and was originally obtained from Dr. Feldmann, Rocky Mountain Laboratories, Hamilton, MT, USA [21].

### 2.2. Preparation of Culture Supernatant and Cell Pellets

Eight T225 mL tissue culture flasks (Corning, Corning, NY, USA) each were infected with EBOV-GFP or SARS-CoV-2-GFP, both recombinant but fully infectious viruses that cause cytopathic effects (CPE), where cells detach from the flask surface and die. Before death, the cells express virus-encoded fluorescent proteins as a marker of infection. Together, these two sensitive measurements were used for detection of infection. Vero E6 cells were grown to 50% confluency in Dulbecco’s modified Eagle Medium (DMEM, Thermo-Fisher, Waltham, MA, USA) supplemented with 10% fetal bovine serum (FBS) supplemented with streptomycin/penicillin (Thermo-Fisher) and then infected with either virus type. The culture was incubated until the entire cell monolayer showed GFP fluorescence (4–5 days post inoculation,) indicating complete infection of cells. The supernatant was collected and placed into 2.0 mL tubes in 1.5 mL aliquots and stored at −80 °C for later use. The virus titers were determined to be 1.3 ± 0.3 × 10^6^ and 2.1 ± 0.2 × 10^6^ focus forming units per mL (FFU/mL) for EBOV and SARS-CoV-2 respectively. The cells were collected using a rubber cell scraper into 10 mL of DMEM. The cells were pelleted at 1000× *g* for 15 min at 4 °C, resuspended to 5 mL in serum free DMEM and aliquoted in 0.22 mL aliquots in tubes and stored frozen at −80 °C. The number of cells in each pellet was counted using a hemocytometer and was 1 × 10^7^. All samples were performed in triplicate with at least two independent experiments performed.

### 2.3. Radiation Treatment

A J.L. Shepherd Model 484R Cobalt-60 irradiator was used. The irradiator delivers gamma radiation from three cobalt 60 sources to a rotating platform to provide even irradiation of samples. It was calibrated by the vendor and confirmatory measurements were taken annually. Measurements were last performed immediately before the present study and was 0.94 MRad per 45 min and within the manufacturer’s specifications (±5%). For this study, times ranging up to 220 min (4.5 MRad) were used.

Samples were aliquoted into 2.0 mL O-ring sealed tubes. The tubes were immersed in at least 5 mL of 10% Microchem Plus (NCL, Philadelphia, PA, USA) in heat sealable pouches/bags (VWR, Radnor, PA, USA) to give at least equal volume to the sample volume in all the tubes. This was done so that in the unlikely event a tube was to leak, the material would be diluted into the 2× disinfectant and become chemically inactivated (minimum concentration of 5% is required). The bag was then heat sealed and stored frozen at −80 °C until day of radiation treatment.

Samples were arranged in a cylindrical container flanking the walls and filled with dry ice. This ensured even sample radiation as the platform rotates. The samples were irradiated and after sufficient time to obtain the desired dose, the samples were removed. The material was returned to the biocontainment lab frozen and stored at −80 °C until time of assay for virus.

Treatments were performed with 3 replicate tubes for 30, 45, 90, 120 and 180 min corresponding to 0.65, 0.97, 1.88, 2.61 and 3.91 MRad respectively. Another independent study evaluated 220 min (4.78 MRad). Samples that remained unexposed to the radiation were treated and stored identically.

### 2.4. Evaluation of Inactivation

The entire content of each vial was added to separate T175 flasks containing Vero E6 cells at 30–40% confluency in 35 mL of DMEM medium supplemented with 2% FBS. This cell density and culture conditions provides optimal growth conditions for each virus. For the cell pellets, virus was released from cells by lysis with 3 freeze thaw cycles. Previous work by our lab and others have shown that freeze-thawing cells efficiently releases viable virus [22]. The flasks were incubated for up to 13 days. Culture supernatants were titrated to calculate the log-kill relationship of the radiation dose. Another set of flasks received no virus. After 7 days, 0.5 mL of each flask was passed in duplicate (total of 1 mL) onto wells of a 6-well plate and incubated for an additional 7 days. Images were taken for the primary flasks up to day 13 and secondary passages up to day 7 by fluorescence and transmitted light microscopy. This two passage approach, based on burst sizes for coronaviruses and EBOV, of about 500 viable viruses per cell over an infected cell lifetime [23,24] indicates we would be able to detect a single viable virus by generalized CPE of the cell monolayer on day 4–5 and fluorescence at earlier time points.

### 2.5. Cytometric Bead Array Assay

Samples in cell culture media containing known and unknown concentrations of cytokines were analyzed using cytometric bead array (CBA) assays. RPMI media supplemented with 10% FBS, 1% penicillin/streptomycin, 1% HEPES and 0.1% BME was spiked with 20 pg/mL of human IL-2 (R&D systems #020-IL-050) or IL-12p70 (PeproTech, #210-12–10 μg, Cranbury, NJ, USA) to generate a sample set with a known cytokine concentration. The unknown sample set was derived from an in vitro T regulatory (Treg) cell suppression assay, where murine Treg cells and T conventional (Tcon) cells were co-cultured to determine the suppressive capacity of Treg cells, resulting in the generation proinflammatory cytokines. The supernatants from this assay were collected and frozen prior to analysis. Known and unknown samples (0.5 mL) were subjected to indicated doses of gamma-irradiation or were left untreated. Following irradiation, known samples plated in quadruplicate were assessed for human IL-2 or IL-12p70 levels using the Flexset (BD#558303) and CBA Human IL-2 Flex Set (#558270) respectively. Unknown samples plated in quadruplicate were assessed for TNF, IFN-gamma, IL-6, IL-10 and CCL2 analyzed on a Beckman CytoFLEX Flow Cytometer. Mean fluorescence intensity (MFI) for each replicate was quantified using FlowJo software and standard cures were generated from MFI values to calculate the concentration of cytokines per sample in pg/mL according to BD CBA protocols.

## 3. Results

### Comparison of CPE and Fluorescent Protein Expression for Detection of Infection

We first established the timing of fluorescent protein expression (GFP or mNeonGreen) and cytopathic effects (CPE) that would be used to detect viable virus. GFP expression was seen as early as 24 h after infection (MOI 0.01) with both viruses and was clearly visible by day 2. As anticipated, CPE was clearly visible by day 2–3 for SARS-CoV-2 and by day 5–6 for EBOV (Figure 1). Some regrowth of cells had occurred by day 7 for the SARS-CoV-2 infected cultures. Culture supernatants lacking virus but treated with gamma radiation did not appear to cause CPE (not shown) indicating that any free radicals had dissipated to non-cytotoxic levels. Overall, a strong correlation was seen between the expression of the fluorescent protein and CPE with fluorescent protein expression always followed by CPE, making fluorescence a reliable marker for the detection of viable virus.

To assess the impact of irradiation on virus inactivation, individual aliquots of virus in supernatant or cell pellets were placed into the irradiator. The placement of tubes within the irradiator was done to ensure equal exposure to the Cobalt 60 source (Figure 2A). Samples were then exposed to radiation for increasing periods of time with source output being constant. Times of 45, 90 and 180 and 220 min, corresponding to 0.97, 1.88, 3.91 and 4.78 MRad, respectively, were used for treatment of supernatants and cell pellets of both viruses. For EBOV alone, additional doses of 0.65 and 2.6 MRad were also tested. During processing and to aid in virus release from cells, each vial was freeze thawed 3 times. The amount of viable virus released from 3 replicate batches of cells was 1.1 × 10^7^ ± 9.8 × 10^6^ and 5.8 × 10^5^ ± 1.2 × 10^5^ per 10^7^ cells that made up the pellet for EBOV and SARS-CoV-2 respectively.

Remaining viable virus was assessed by placing irradiated material on monolayers of VeroE6 cells and incubating the mixture for 13 days, and additionally by passaging the supernatant twice as described in Figure 2B. SARS-CoV-2 viability was greatly reduced after treatment of supernatant or cell pellets with the lowest level of radiation tested, 0.9 MRad. Cell supernatants and cell pellets still showed evidence of GFP positive cells and CPE but was delayed by several days and was comparable to a 1:100 dilution of the original virus inoculum. This indicated that 0.9 MRad significantly reduced virus titer by approximately 2 logs but did not fully inactivate supernatant and pellet samples. Additional treatment of SARS-CoV-2 infected supernatant or cell pellets with a dose of 1.9 MRad or 3.8 MRad showed no viable virus in both the supernatant and cell pellet material (Figure 3) demonstrating complete inactivation at 1.9 MRad and above.

Unlike SARS-CoV-2, EBOV required higher doses of radiation for complete inactivation (Figure 4). While readouts from day 13 cultures and second passage day 7 cultures were in agreement, passaging of virus appeared better able to detect very low levels of residual virus as the virus was able to replicate better on fresh cells and we estimate to be be able to detect as low as single viable virus particles (Figure 4). When compared to the untreated control, the virus titer for the 0.65 and 0.97 MRad doses was reduced by 238 and 367-fold. The calculation of a log-kill relationship [11] yielded 3.1 log-fold virus titer reduction per MRad (Figure 5). Based on this relationship, we expected to have little to no virus present for the 2.6 MRad treatments (calculated to reduce titer by greater than 2 × 10^6^-fold and total starting virus amount in the 1.5 mL sample was calculated as 1.95 × 10^6^ FFU). Indeed, for supernatants containing EBOV, no evidence of viable virus was present at this dose. However, the cell pellets showed sporadic infection, with isolated patches of infection being observed in two of three replicates for day 13 flasks and second pass cultures. Higher doses of radiation (3.8 MRad) rendered the cell pellets non-infectious with no evidence of replication from all flasks.

Based on the relatively high doses of radiation needed to inactivate filoviruses, we wanted to better understand how treatment might affect the ability to detect proteins, in particular, proinflammatory cytokines that are known to be secreted by cells during virus infection. The proinflammatory cytokines CCL2, IL-6, TNF, IFN-gamma and IL-10 have each been detected in cells infected by numerous virus types including coronaviruses and filoviruses [17] and are seen in infected patients [25]. Cell culture supernatant from an in vitro murine T regulatory (Treg) cell assay, where murine Treg cells and T conventional (Tcon) cells were co-cultured, was used to generate a mixture of these proinflammatory cytokines at unknown concentrations. We also prepared culture medium containing physiologically relevant levels of human IL-2 or IL-12p70 by addition of each to a final concentration of 20 pg/mL.

Each sample set was exposed to 1.9 or 2.6 MRad of gamma radiation and cytokine concentrations were quantified using cytometric bead array (CBA) assays. While the radiation dose consistently reduced the level of each cytokine detected, the drop was relatively small, remaining at 82 ± 7% of the initial cytokine levels, with only small differences seen after treatment with either radiation dose (Figure 6). Furthermore, each of the cytokines was impacted similarly despite having different amino acid compositions and structures. Overall, while the radiation doses used inactivated each virus, they had only a minor effect on detection of cytokines indicating that treatment will preserve such markers of infection.

## 4. Discussion

Our work shows that a dose of 2.6 MRad was sufficient to render culture supernatants containing 2 × 10^6^ FFU of EBOV completely inactivated. At this dose, EBOV in cell pellets still had residual infectivity and required a longer treatment of up to 3.8 MRad for complete inactivation. The dose needed for EBOV was similar to that reported in another study using the same irradiator model, where 2 MRad inactivated 10^6^ PFU of Ebola virus [12]. An earlier report indicated that 1.3 MRad was sufficient for a higher virus load of 10^6^ TCID50. For EBOV, TCID50 titer measurements are typically less sensitive (5–10 fold) compared to FFU assays [26] indicating that the virus load was up to 10-fold higher than that tested here but with 2-fold less radiation needed to achieve inactivation. This difference is likely greater than inaccuracies in virus titration assays and is likely due to differences in the design of the irradiator or how samples were positioned in proximity to the radiation source. Unfortunately, information on this older irradiator model made in 1977 (model 220, Atomic Energy of Canada) was not readily available. Other differences known to affect inactivation efficiency, such as serum concentration and temperature, were the same, with each study having 10% serum present and performed using dry ice to maintain samples frozen [11].

Compared to EBOV, SARS-CoV-2 was more susceptible to inactivation, with 1.9 MRad completely inactivating supernatants containing 3 × 10^6^ FFU SARS-CoV-2. This dose is similar to that reported to inactivate a distantly related coronavirus, MERS-CoV, where 2 MRad was sufficient for inactivation of a similar amount of virus in a similar model irradiator, JL Shepherd 484R2 [16]. This consistent outcome indicates that, in general, we expect our findings to apply to other coronaviruses as their composition is essentially the same as SARS-CoV-2 and MERS-CoV. The higher sensitivity of SARS-CoV-2 to inactivation compared to EBOV may reflect the larger size of its genome, the way the genome is packaged in the virion, or other compositional differences. Given that filoviruses all share a similar morphology and composition to each other, it is expected that each would require similarly high radiation doses to that needed to inactivate EBOV. However, if samples contain higher virus amounts, they would need to be diluted in buffer or medium to match conditions used here or independently verified.

To our knowledge, inactivation of virus in cell pellets has not been previously evaluated. This is potentially useful for proteomic work to study virus-host protein interactions or for therapy development or diagnostics where cell responses to infection need to be measured. Similarly, measuring cytokine levels in cell cultures, animal disease models, or patients is important to understand the infection process and the immune response for diagnostic and therapy development. Exposure to inactivating chemicals such as formalin, or denaturants such as guanidinium salts or SDS, often prevent follow up biological assays. Here, we used cytokine measurement to gauge the impact of irradiation on protein folding and detection. It was encouraging to see that for all cytokines tested the radiation treatment caused relatively small but consistent decreases (on average 15%) in the amount of cytokine able to be detected. Given that larger changes in cytokine levels (>2-fold changes are seen in EBOV infected macaques) would be seen during a proinflammatory response in infected cells [27], the impact of radiation on measurements is relatively small and given the consistent change seen, original levels could be back calculated. This finding opens up the potential to use gamma radiation for cell culture supernatants as well as animal serum for the study of in vitro and in vivo responses to infection.

Protein content of the mixture being treated can affect inactivation efficiency. Higher levels of serum reduce inactivation efficiency, requiring higher doses for complete inactivation [13]. Therefore, undiluted animal serum would either require longer treatment times or serum would need to be diluted to 10% to align with what has been tested here. For cell pellets of 1 × 10^7^ in 0.25 mL of DMEM, no serum was present. However, cell proteins are present and would be, overall, proportional to the cell number. It is likely the concentrated cell proteins, present in cell pellets, resulted in the need for higher radiation dosage for inactivation compared to supernatants (3.8 vs. 2.6 MRad). For sensitivity and consistency, we used Vero E6 cells to culture samples as both EBOV and SARS-CoV-2 grow to higher titers in VeroE6 than in other commonly used cell lines such as HeLa and A549 cells. However, if virus growth in the cells of choice exceeds that of VeroE6 cells, the number of cells used in the cell pellet will need to be reduced to match the viral load present in the Vero cells.

Overall, gamma radiation is an effective tool for inactivation of virus-containing samples, allowing samples to be further evaluated outside of high biocontainment laboratories. The process is highly reproducible, as demonstrated by several reports, but may be affected by the design of the irradiator. Care should be taken to optimize inactivation parameters for the particular virus quantity and protein content of the sample. Importantly, while gamma irradiation is disruptive for virus replication, likely due to fragmentation of the virus genome and denaturation of virion components, individual proteins such as cytokines survive irradiation and can be assayed using standard assays. However, for larger macromolecules, degradation will be more likely and should be taken into account when assaying treated material. With these caveats, while not evaluated here, successful inactivation of cell pellets will enable further evaluation of other proteins involved in virus infection, such as host factors used by viruses during replication, or those involved in host responses to infection such as the innate immune response.

## Figures and Tables

**Figure 1 viruses-15-00043-f001:**
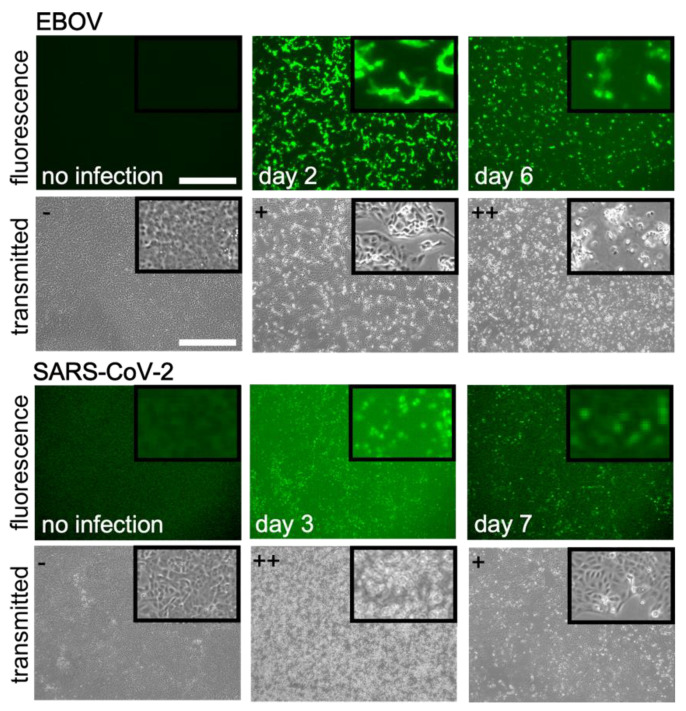
Timing and detection of virus spread. Fluorescence and transmitted light images of EBOV and SARS-CoV-2 infected Vero E6 cells on indicated days after infection. Images were taken on the indicated days when green fluorescence or CPE was visible and compared to cells without virus. GFP (EBOV) or mNeonGreen (SARS-CoV-2) signals precede CPE for both viruses. For EBOV, GFP is strongly expressed on day 2. For SARS-CoV-2, the signal was fainter but clearly visible on day 3 together with CPE. For EBOV, CPE appeared later than SARS-CoV-2 being evident on day 7. At top left of brightfield images grading for CPE is given with − = none, + = low and ++ = high CPE. Scale bar is 500 mm with all images taken at the same magnification.

**Figure 2 viruses-15-00043-f002:**
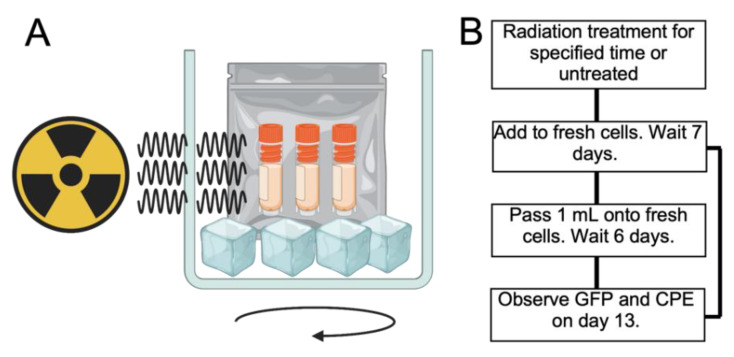
Schematic showing arrangement of tubes in gamma radiator and workflow. (**A**) Tubes containing virus infected cell lysates or culture supernatants containing virus were sealed in plastic bags containing 10% Microchem Plus and frozen on dry ice. The samples were placed in a cylindrical container which was placed on a rotating platform. Three radiation sources were present to provide even coverage of samples. (**B**) Schematic showing approach used. After tubes were irradiated with the desired dose, they were returned to the high containment laboratory and added to flasks of fresh cells. These were incubated for 7 days and then passed onto fresh cells or allowed to further incubate for up to 13 days.

**Figure 3 viruses-15-00043-f003:**
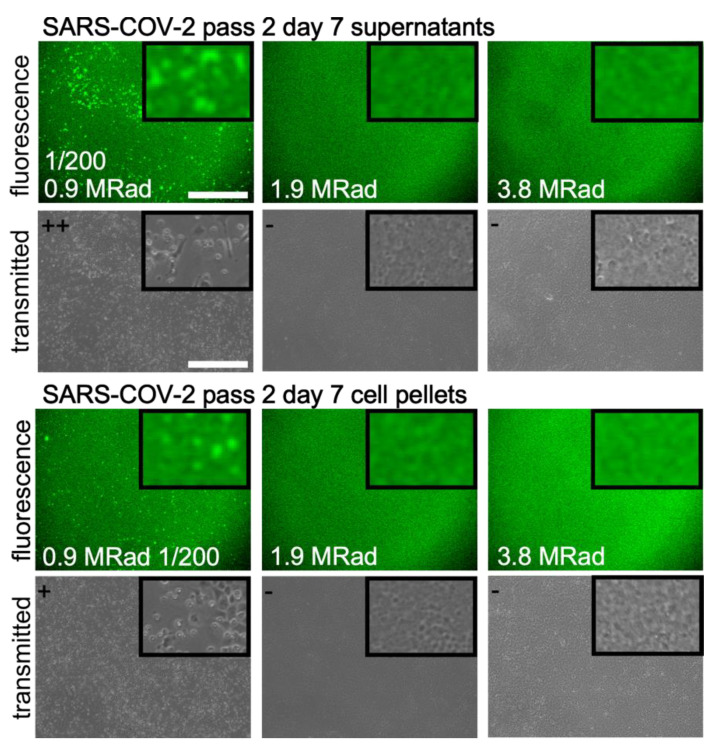
Inactivation of SARS-CoV-2 by gamma radiation. Supernatants were collected from the primary culture flasks after 7 days post infection and used to inoculate fresh cells. Images were taken after an additional 7 days. For the 0.9 MRad dose, undiluted samples caused excessive cell death by day 7 and poor image quality. So, an image for culture medium diluted 1/200 is shown for this dose only with other samples being undiluted. Insets are 5× magnification of the parent image. At top left of brightfield images grading for CPE is given with − = none, + = low and ++ = high CPE. Scale bar is 500 mm with all images taken at the same magnification.

**Figure 4 viruses-15-00043-f004:**
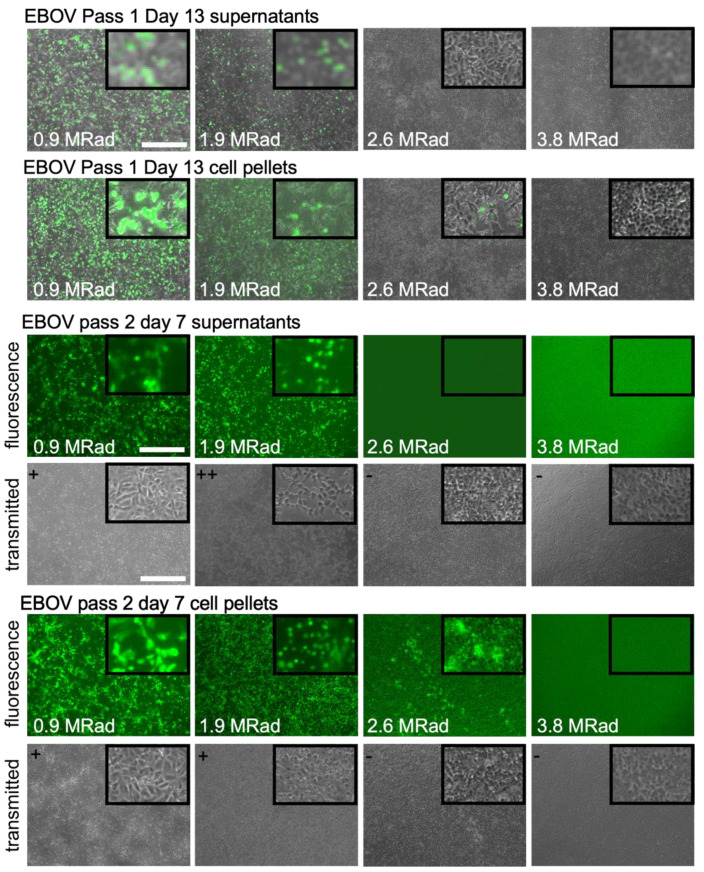
Inactivation of EBOV by gamma radiation. Images (bright field and fluorescence) were taken at the indicated times after irradiated material was used to inoculate flasks of fresh Vero cell monolayers and incubated for 13 days or passed onto fresh cells after 7 days as indicated. Both cell pellets and culture supernatants were evaluated for inactivation. Insets show isolated regions where infection was detected and are 5× magnification of the parent image. At top left of brightfield images grading for CPE is given with − = none, + = low and ++ = high CPE. Scale bar is 500 mm with all images taken at the same magnification.

**Figure 5 viruses-15-00043-f005:**
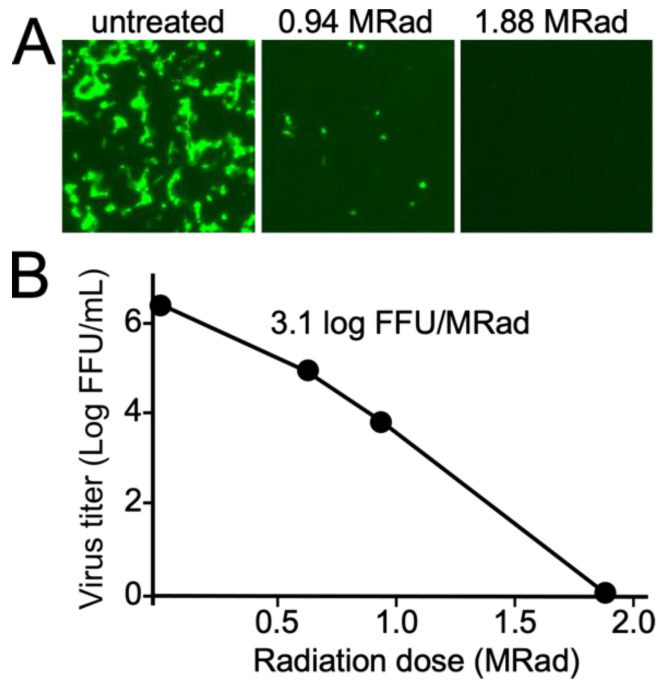
Calculation of radiation dose:log kill relationship for Ebola virus. Samples were treated with the indicated levels of gamma radiation. The amount of viable virus remaining was calculated by titration of samples in 96-well plates. Foci expressing green fluorescent protein were counted and used to calculate the amount of virus in the sample. (**A**) Representative images of infected cells are shown after 2 d of incubation on Vero cells. (**B**) Plot of amount of viable virus present vs. radiation dose. The indicated dose:virus kill relationship was determined by fitting a line equation of y = mx + c to the indicated log virus titer vs. radiation dose data in Excel.

**Figure 6 viruses-15-00043-f006:**
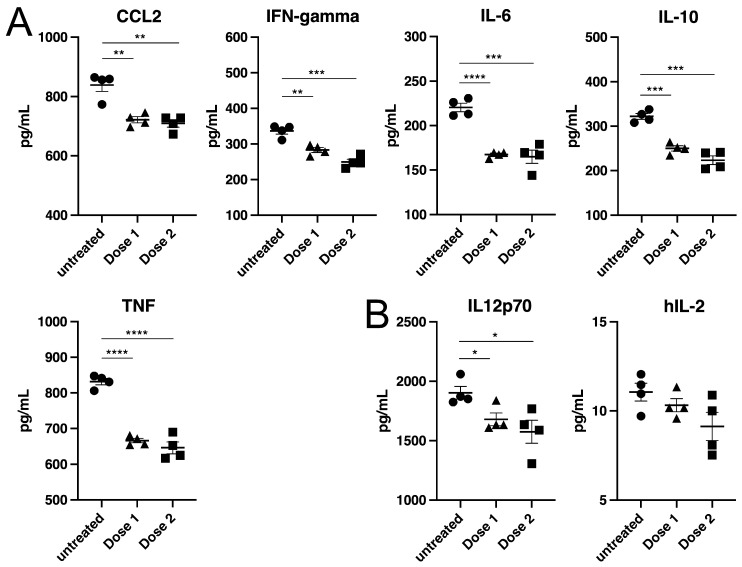
Measurement of cytokines after irradiation of medium. (**A**) A mixture of murine proinflammatory cytokines, produced from interaction of murine Tcon and Treg cells, was used to evaluate impact on cytokine detection (**B**) The amount of human IL-12p70 and IL-2 remaining in culture medium after irradiation. Samples were dosed at 1.9 (Dose 1) or 2.6 MRad (Dose 2). Amounts of the indicated cytokines were then measured using cytometric bead array assays. Replicates are indicated together with means and standard deviations. Circles are for untreated samples, triangles for 1.9 MRad and squares for 2.6 MRad dosed samples. * *p* < 0.05, ** *p* < 0.01, *** *p* < 0.001, **** *p* < 0.0001 using Student’s unpaired *t*-test to compare untreated to treated samples.

## Data Availability

All data is contained within this article.

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
