# Peer review of "Inactivation of Ebola Virus and SARS-CoV-2 in Cell Culture Supernatants and Cell Pellets by Gamma Irradiation"

_viruses, 2022, doi:10.3390/v15010043_

Round 1
Reviewer 1 Report
This is a well written paper that presents some novel data on the effect of gamma irradiation on EBOV and SARS-CoV-2 cell pellets. It also presents further information on irradiation on supernatants.
My main comments are related to the results (see below).
Minor comments
Introduction
-References or more information needed lines 52-57 and 74-76>
Methods
-Lines 79-86 appear to be in a different font than the rest of the text
-Spell out DMEM and FBS? (line 93)
-Centrifugation details needed (line 100)
-How was the number of cells in the pellet determined? (line 102)
-Manufacturer/supplier details missing throughout
-Lines 111-116 - was the immersion in bags containing MicroChem done at the same time as harvest (line 97) or on a different occasion? I am unclear how many times samples were freeze-thawed prior to irradiation, which may affect titer.
-What was the titer of the control samples mentioned in lines 124-125? These are not mentioned again but are an important control.
-Throughout but first mentioned lines 145 and 152 - is it human IL-12 or IL-12p70?
Results
-MAJOR COMMENT - I think all figures with cell images need improving. I think each panel (not just Figure 1) should include a no infection image, and also a positive control infection image (the control samples mentioned in lines 124-125 that were not irradiated but went through same freeze thawing. It might be down to my printer and eyesight but I do not think the images are that clear. I would suggest adding a '+' or a '-' to each image to indicate whether it is infected or not. In Figure 1, SARS-CoV-2 transmitted, no infection and day 7 look similar and image does not clearly match with text lines 164-165. Figure 1 legend also lacks description of the inset. Figure 4, green looks different for Pass 1 images compared to Pass 2 - have images been manipulated? Do all figures need a scale? Figure 4, 2.6 MRad label missing for pass 2 day 7 supernatant.
-Line 195 - please explain how 238- and 367-fold was reached and how kill curve in Figure 5 was determined
-Line 199 - please explain titer of 1.95 (not the same titer as given in line 97)
-Lines 210-215 appear to be in a different font than the rest of the text
-Figure 6 legend missing what one * asterix means (p for ** to **** listed( but IL12(p70) graph only has one *
Author Response
Responses to the reviewer comments are indicated by the bullet dot.
Minor comments
Introduction
-References or more information needed lines 52-57 and 74-76>
- Added 2 references numbered as 14 and 15. One regarding radiation damage to macromolecules and the other on cytokine production after virus infection.
Methods
-Lines 79-86 appear to be in a different font than the rest of the text
- This is corrected
-Spell out DMEM and FBS? (line 93)
- This is done.
-Centrifugation details needed (line 100)
- Added 15 minutes 4°C to line 102.
-How was the number of cells in the pellet determined? (line 102)
- A hemocytometer was used and this is added on line 104.
-Manufacturer/supplier details missing throughout
- Added this information at multiple points.
-Lines 111-116 - was the immersion in bags containing MicroChem done at the same time as harvest (line 97) or on a different occasion? I am unclear how many times samples were freeze-thawed prior to irradiation, which may affect titer.
- This is an important point. It was 3 times. This is now indicated in the methods on line 134-136 and is now repeated in the results section on line 182-183.
-What was the titer of the control samples mentioned in lines 124-125? These are not mentioned again but are an important control.
- The titers were assessed in 3 replicates and are now given on lines 183-185
-Throughout but first mentioned lines 145 and 152 - is it human IL-12 or IL-12p70?
- Recombinant mouse IL-12p70 was used which is the active form of IL-12. We have corrected the manuscript to reflect this and added supplier and catalog numbers in the methods on lines 149-150. Assay kit catalog numbers were added on lines 157-158.
Results
-MAJOR COMMENT - I think all figures with cell images need improving. I think each panel (not just Figure 1) should include a no infection image, and also a positive control infection image (the control samples mentioned in lines 124-125 that were not irradiated but went through same freeze thawing. It might be down to my printer and eyesight but I do not think the images are that clear. I would suggest adding a '+' or a '-' to each image to indicate whether it is infected or not. In Figure 1, SARS-CoV-2 transmitted, no infection and day 7 look similar and image does not clearly match with text lines 164-165. Figure 1 legend also lacks description of the inset. Figure 4, green looks different for Pass 1 images compared to Pass 2 - have images been manipulated? Do all figures need a scale? Figure 4, 2.6 MRad label missing for pass 2 day 7 supernatant.
- We have modified the figures. A scale bar is now included with each set of images and the figure legend has been updated. The missing label is added. Where possible the resolution of the images has been improved. As suggested, we have included a grading of CPE for the brighfield images to aid the reader in interpreting the outcomes. The pass 1 images look different to the pass 2 images as the pass 1 images are composites of the GFP and brightfield images while pass 2 have the GFP and brightfield split out in separate channels. We did this to conserve space for the figure as the main conclusion is taken from the pass 2 day 7 images.
-Line 195 - please explain how 238- and 367-fold was reached and how kill curve in Figure 5 was determined
- This was calculated by comparing the titers of untreated vs treated samples. We have added this to lines 203-204.
-Line 199 - please explain titer of 1.95 (not the same titer as given in line 97)
- Line 207 shows the total amount of virus in the 1.5 mL sample. We added this to the description of the sample on line 207.
-Lines 210-215 appear to be in a different font than the rest of the text
- Corrected font.
-Figure 6 legend missing what one * asterix means (p for ** to **** listed( but IL12(p70) graph only has one *
- This has been corrected to indicate P<0.05.
Reviewer 2 Report
In their manuscript entitled “Inactivation of Ebola virus and SARS-CoV-2 in Cell Culture Supernatants and Cell Pellets by Gamma Irradiation” Boytz and Co-workers applied various doses of gamma irradiation from a cobalt 60 source toward supernatant and cell pellets from cell cultures infected with recombinant strains of SARS-CoV2 and Ebola Virus. While inactivation studies with gamma irradiation and such viruses new, I understand that such studies may help conducting further research on virus-infected materials outside a BSL3 or BSL-4 facility. On the other hand, the Method described here will require that infectious material is exported from the BSL lab for gamma irradiation.
Major point
- For me , the only scientifically new aspect is that, in this study, frozen cell pellets have been subjected to gamma irradiation, and that Ebola Virus inactivation in cell pellets. was not complete. However, in order to appropriately assess viral inactivation capacity it is common practice to quantify the infectious load before and after inactivation. While the infectious titer was determined from cell culture supernatants before inactivation, such data have not been presented for the frozen pellets before inactivation. Infectious load might be much higher in cell pellets the in supernatant and this could explain the incomplete inactivation observed in this study. Therefore, it is considered essential to complete such data before publication
Additional Points
- Discussion lines 227-233. It should be realised that accuracy of virus quantification using only three replicas and 10 fold dilution is quite low. I understand that more accurate format for virus titration via end point dilutions is associated with workload for such viruses. Accuracy of virus titration from previous publications should also be critically reviewed. The discussion might benefit from some estimation whether such inaccuracy cous explain a log difference in virus inactivation capacity
- The method for quantification of viral titer and calculations leading to Figure 5 should be explained more in detail.
- While high dose gamma irradiation may not affect small molecules such as cytokines, it will destroy large and sensitive molecules such as clotting factors. Such limitations should be outlines in the discussion section.
- Fig 1: Although Figure 1 is a ‘nice-to have’ set of pictures. I do not consider this essential for the manuscript.
- Line 248: It is said that “Filoviruses also have similar morphology and composition to each other and should be inactivated similarly to EBOV”. However, Coronaviruses have a larger single stranded and non-segmented RNA Genome (ca. 30kb) than Filoviruses (18-19kb) which readily explains the higher susceptibility towards gamma irradiation.
Author Response
Responses to review comments are indicated by the bullet dot.
Major point
- For me , the only scientifically new aspect is that, in this study, frozen cell pellets have been subjected to gamma irradiation, and that Ebola Virus inactivation in cell pellets. was not complete. However, in order to appropriately assess viral inactivation capacity it is common practice to quantify the infectious load before and after inactivation. While the infectious titer was determined from cell culture supernatants before inactivation, such data have not been presented for the frozen pellets before inactivation. Infectious load might be much higher in cell pellets the in supernatant and this could explain the incomplete inactivation observed in this study. Therefore, it is considered essential to complete such data before publication
- We have added the titers of virus extracted from cell pellets. Measurements were done for 3 experimental replicates. This is given on lines 184-186.
Additional Points
- Discussion lines 227-233. It should be realised that accuracy of virus quantification using only three replicas and 10 fold dilution is quite low. I understand that more accurate format for virus titration via end point dilutions is associated with workload for such viruses. Accuracy of virus titration from previous publications should also be critically reviewed. The discussion might benefit from some estimation whether such inaccuracy cous explain a log difference in virus inactivation capacity
- This is a good point. While 10-fold dilutions were used, titer was calculated by counting foci and so gained additional accuracy. We have made a comment that the difference is likely to be more than just error in the calculation on lines 245-246.
- The method for quantification of viral titer and calculations leading to Figure 5 should be explained more in detail.
- We have updated the figure legend for Fig. 5 to better describe the method used for calculating the log kill relationship.
- While high dose gamma irradiation may not affect small molecules such as cytokines, it will destroy large and sensitive molecules such as clotting factors. Such limitations should be outlines in the discussion section.
- This is an important point and we have added a comment about larger macromolecules being affected more than small ones. This is on lines 298-300.
- Fig 1: Although Figure 1 is a ‘nice-to have’ set of pictures. I do not consider this essential for the manuscript.
- Figure 1 shows the outcomes when no treatment is given. Reviewer 1 indicated that this was needed.
- Line 248: It is said that “Filoviruses also have similar morphology and composition to each other and should be inactivated similarly to EBOV”. However, Coronaviruses have a larger single stranded and non-segmented RNA Genome (ca. 30kb) than Filoviruses (18-19kb) which readily explains the higher susceptibility towards gamma irradiation.
- This is a good point. We have added this discussed on lines 258-262.

Round 2
Reviewer 2 Report
The points raised in the first review have been sufficientrly addressed
Major Point 1
For me , the only scientifically new aspect is that, in this study, frozen cell pellets have been subjected to gamma irradiation, and that Ebola Virus inactivation in cell pellets. was not complete. However, in order to appropriately assess viral inactivation capacity it is common practice to quantify the infectious load before and after inactivation. While the infectious titer was determined from cell culture supernatants before inactivation, such data have not been presented for the frozen pellets before inactivation. Infectious load might be much higher in cell pellets the in supernatant and this could explain the incomplete inactivation observed in this study. Therefore, it is considered essential to complete such data before publication
Response: We have added the titers of virus extracted from cell pellets. Measurements were done for 3 experimental replicates. This is given on lines 184-186.
Reviewer’s comment:
Point resolved
Additional Points
Point 1
Discussion lines 227-233. It should be realised that accuracy of virus quantification using only three replicas and 10 fold dilution is quite low. I understand that more accurate format for virus titration via end point dilutions is associated with workload for such viruses. Accuracy of virus titration from previous publications should also be critically reviewed. The discussion might benefit from some estimation whether such inaccuracy cous explain a log difference in virus inactivation capacity
Response This is a good point. While 10-fold dilutions were used, titer was calculated by counting foci and so gained additional accuracy. We have made a comment that the difference is likely to be more than just error in the calculation on lines 245-246.
Reviewer’s comment:
Point resolved
Point 2
The method for quantification of viral titer and calculations leading to Figure 5 should be explained more in detail.
Response: We have updated the figure legend for Fig. 5 to better describe the method used for calculating the log kill relationship.
Reviewer’s comment:
Point resolved
Point 3
While high dose gamma irradiation may not affect small molecules such as cytokines, it will destroy large and sensitive molecules such as clotting factors. Such limitations should be outlines in the discussion section.
Response: This is an important point and we have added a comment about larger macromolecules being affected more than small ones. This is on lines 298-300.
Reviewer’s comment:
Point resolved
Point 4:
Fig 1: Although Figure 1 is a ‘nice-to have’ set of pictures. I do not consider this essential for the manuscript.
Response: Figure 1 shows the outcomes when no treatment is given. Reviewer 1 indicated that this was needed.
Reviewer’s comment:
acceptable
Point 5
Line 248: It is said that “Filoviruses also have similar morphology and composition to each other and should be inactivated similarly to EBOV”. However, Coronaviruses have a larger single stranded and non-segmented RNA Genome (ca. 30kb) than Filoviruses (18-19kb) which readily explains the higher susceptibility towards gamma irradiation.
Response: This is a good point. We have added this discussed on lines 258-262.
Reviewer’s comment:
Point resolved